# Obesity and Dysmetabolic Factors among Deceased COVID-19 Adults under 65 Years of Age in Italy: A Retrospective Case-Control Study

**DOI:** 10.3390/v14091981

**Published:** 2022-09-07

**Authors:** Loreta A. Kondili, Maria Giovanna Quaranta, Mauro Viganò, Xhimi Tata, Franca D’Angelo, Cinzia Lo Noce, Luigi Palmieri, Graziano Onder, Federico D’Amico, Elvira Inglese, Massimo Puoti, Alessio Aghemo, Maria Elena Tosti

**Affiliations:** 1Center for Global Health, Istituto Superiore di Sanità, 00161 Rome, Italy; 2Hepatology Unit, San Giuseppe Hospital, 20123 Milan, Italy; 3Department of Cardiovascular, Endocrine-Metabolic Diseases and Aging, Istituto Superiore di Sanità, 00161 Rome, Italy; 4Infectious Disease Unit, ASST Grande Ospedale Metropolitano Niguarda, 20162 Milan, Italy; 5Department of Brain and Behavioral Sciences, University of Pavia, 27100 Pavia, Italy; 6Clinical Chemistry Laboratory, ASST Grande Ospedale Metropolitano Niguarda, 20162 Milan, Italy; 7School of Medicine, Università degli Studi di Milano-Bicocca, 20126 Milan, Italy; 8Department of Gastroenterology, IRCCS Humanitas Research Hospital IRCCS, 20089 Rozzano, Italy; 9Department of Biomedical Sciences, Humanitas University, 20072 Pieve Emanuele, Italy

**Keywords:** metabolic impairment, young adults, risk factors, clinical outcome

## Abstract

Background: Italy has witnessed high levels of COVID-19 deaths, mainly at the elderly age. We assessed the comorbidity and the biochemical profiles of consecutive patients ≤65 years of age to identify a potential risk profile for death. Methods: We retrospectively analyzed clinical data from consecutive hospitalized-for-COVID-19 patients ≤65 years, who were died (593 patients) or discharged (912 patients) during February–December 2020. Multivariate logistic regression identified the mortality risk factors. Results: Overweight (adjusted odds ratio (adjOR) 5.53, 95% CI 2.07–14.76), obesity (adjOR 8.58, CI 3.30–22.29), dyslipidemia (adjOR 10.02, 95% CI 1.06–94.22), heart disease (adjOR 17.68, 95% CI 3.80–82.18), cancer (adjOR 13.28, 95% CI 4.25–41.51) and male sex (adjOR 5.24, 95% CI 2.30–11.94) were associated with death risk in the youngest population. In the older population (46-65 years of age), the overweight and obesity were also associated with the death risk, however at a lower extent: the adjORs varyied from 1.49 to 2.36 for overweight patients and from 3.00 to 4.07 for obese patients. Diabetes was independently associated with death only in these older patients. Conclusion: Overweight, obesity and dyslipidemia had a pivotal role in increasing young individuals’ death risk. Their presence should be carefully evaluated for prevention and/or prompt management of SARS-CoV2 infection in such high-risk patients to avoid the worst outcomes.

## 1. Introduction

From the outset of the COVID-19 pandemic, accumulating evidence about the degree to which age and underlying morbidities may amplify the severity of infection with SARS-CoV2 has been reported in different population settings and geographical areas around the world. Older age groups have featured prominently in COVID-19-related morbidity and mortality [1,2]. In China, older age and comorbidities, including diabetes, hypertension and cardiovascular and chronic respiratory diseases, were the most prominent high-risk characteristics [3,4,5]. Severe obesity (body mass index (BMI) ≥ 40 kg/m^2^) at any age is a high-risk condition for hospitalization and the Intensive Unit Care for COVID-19 [6]. In the United States, obesity is emerging as an important risk factor. More than 900,000 adult COVID-19 hospitalizations occurred in the United States between the beginning of the pandemic and 18 November 2020. Models estimate that 271,800 (30.2%) of these hospitalizations were attributed to obesity [7]. Obesity could shift severe COVID-19 disease to younger ages [8]. However, how obesity and, potentially, metabolic syndrome factors could affect the infection’s outcomes in different age groups and the weight of each of them in younger compared with the elderly age are still debated questions.

Strong recommendations are given for the use of booster COVID-19 vaccines, as well as antivirals after the SARS-CoV2 infection in the elderly, considered a fragile population at risk for the worst COVID-19 outcomes [9]. However, it is also important to understand better the risk factors that predispose patients not at elderly age to COVID-19 severe outcomes and high-risk of death. To this end, we aimed to retrospectively evaluate the comorbidities and risk factors independently associated with the death among COVID-19-positive hospitalized patients under 65 years of age (<65 years) in Italy. The final goal is to help in defining the young population that could be considered at high risk to be addressed for vaccine boosters and prompt antiviral use following SARS-CoV2 infection.

## 2. Methods

### 2.1. Study Population and Data Collection

At the outset of the COVID-19 outbreak, the Istituto Superiore di Sanità (ISS) launched an integrated national surveillance system (the ISS COVID-19 surveillance system) to collect information on individuals with COVID-19 from all Italian regions.

The study population consisted of a random sample of 593 individuals ≤65 years available in the ISS COVID-19 surveillance system, who died in Italian hospitals with confirmed COVID-19, between February 2020 and December 2020. COVID-19-related deaths were defined as those occurring in patients who tested positive for SARS-CoV2 through reverse transcription polymerase chain reaction, independently of pre-existing diseases that may have caused or contributed to the death [10,11]. Deceased patients were compared with a random sample of 914 consecutive patients aged ≤65 years, hospitalized due to COVID-19 at the ASST Grande Ospedale Metropolitano Niguarda in Italy, improved and discharged alive during the same period.

Laboratory parameters for each discharged alive patient were retrospectively extracted from a panel of test results performed in clinical routine. The tested chemical analytes included: Aspartate Transaminase (AST), Alanine Aminotransferase (ALT), Triglycerides, Glycemia, Bilirubin, Lactate Dehydrogenase (LDH), C-reactive protein (CRP) and Creatinine. Patients’ coagulation status was monitored using the International Normalized Ratio (INR) and D-Dimer (DD) plasma levels. To obtain laboratory variables, a query was created to extract anonymized data using the patients’ ID (a numeric string) from an SQL-based repository in which all analytical results of the tests performed in the laboratory were stored. The fields extracted were sex, date of birth, day of lab tests execution, tests ID, test results and hospital ward.

Medical charts of COVID-19 patients dying in hospital were reviewed by a group of researchers at ISS and demographic, comorbidity and biochemical data were extracted and recorded in a dedicated electronic Case Report Form (eCRF) and a specific data base with all the items collected according to the study design was created. The laboratory parameters were evaluated normalizing each parameter for the value considered as the reference. Data on the following comorbidities: obesity, diabetes, dyslipidemia, hearth disease, hypertension, chronic obstructive pulmonary disease (COPD), cancer, chronic kidney disease (CKD) and chronic liver disease were collected. The Fibrosis-4 (Fib-4) index, as a non-invasive estimate of liver fibrosis, was generated in the eCRF.

### 2.2. Statistical Analysis

Patients’ main characteristics were reported as median and range or as proportions (N and %) for continuous and categorical variables, respectively. The Mann–Whitney U test was used for the continuous variables to assess differences between the distributions and the Chi-squared test (or F-Fisher test, if necessary) was used for the comparison of the proportions. Non-parametrical tests were used to be more conservative in defining the statistical significance in particular in lack of power when the normality is not ensured. A *p*-value of <0.05 was considered statistically significant.

Variables independently associated to death were evaluated by multivariate logistic models in the overall population and for specific age groups: 20–45, 46–55 and 56–65 years of age. The following variables were included in the model: age, gender, overweight/obesity, diabetes, dyslipidemia, hearth disease, hypertension, COPD, cancer and CKD. Regarding the use of the variable “age”, in each model it was introduced as continuous as the effect of age on COVID-19 lethality is linear. All analyses were performed using the STATA/SE 16.1 statistical package (StataCorp LP, College Station, TX, USA).

### 2.3. Ethics

This study was carried out in keeping with the principles of the Declaration of Helsinki. On 27 February 2020, the Italian Government authorized the collection and scientific dissemination of data concerning the COVID-19 epidemic by the ISS and other public health institutions [12].

## 3. Results

### 3.1. Clinical Characteristics of the Study Population

Both deceased and discharged patients had a similar median hospitalization time of 11 days [with interquartile range (IQR) 6–22 days and 7–18 days, respectively; *p* = 0.737]. The demographic and comorbidity patterns of deceased hospitalized patients and those who survived are shown in Table 1.

Among the 593 deceased individuals, 460 (77.6%) were male, while the majority of discharged patients were female (N = 611, 66.9%). The median age of deceased and discharged patients was 59 years (range: 22–65) and 52 years (range: 20–65), respectively (*p* < 0.001).

Deceased patients were equally distributed between normal/underweight, overweight and obese, being about one-third per group while the normal/underweight group was prevalent in the discharged individuals (66%). Obesity grade I (BMI ≥ 30 and <35 Kg/m^2^) was prevalent in the survivors (69%), while 54% of the deceased individuals reported grade II (BMI ≥ 35 and <40 Kg/m^2^) or grade III (BMI ≥ 40 Kg/m^2^) obesity (data not shown).

Compared with discharged individuals, deceased patients presented a higher rate of associated comorbidities: diabetes (34.2% vs. 13.9%, *p* < 0.001), dyslipidemia (10.5% vs. 2.5%), heart disease (23.8% vs. 7.5%), hypertension (45.9% vs. 15.7%), COPD (11.3% vs. 3.2%) and CKD (14.3% vs. 4.8%). No difference in the prevalence of ongoing cancer among the two groups was observed (*p* = 0.077).

Overt liver disease was reported in 37 (6.2%) and in 38 (4.2%) deceased and discharged individuals, respectively (*p* = 0.069). Overall, 5.6% deceased patients and 1.1% discharged patients were classified as having advanced fibrosis/cirrhosis by FIB-4 calculation (*p* < 0.001) (data not shown).

A total of 315 (74.9%) deceased individuals vs 358 (39.2%) survivors reported at least one comorbidity related to metabolic impairment (including diabetes, dyslipidemia, heart disease, hypertension and obesity) (*p* < 0.001). More than three comorbidities related to metabolic impairment were reported in 22.6% and in 2.2% of deceased and discharged patients, respectively (*p* < 0.001) (data not shown). Figure 1 shows an increase of the number of comorbidities related to the metabolic impairment in the deceased patients, whereas, on the contrary, a decreased pattern was observed within the discharged individuals.

Considering that the comorbidity profile differs in relation to the age, it was stratified by age and the results are reported in Table 2.

In the youngest age group (25–45 years of age), heart disease, overweight, obesity and cancer were significantly more frequent in deceased vs. discharged individuals, whereas in the oldest age, all comorbidities, but cancer, were more frequently present in patients who did not survive.

The presence of one or more surrogate marker of potential metabolic syndrome (i.e., diabetes, dyslipidemia, heart disease, hypertension and obesity) prevails in the deceased patients in all age groups. In the oldest age group (56–65 years), an increasing trend of more than one surrogate marker of potential metabolic syndrome was observed.

No differences in the comorbidities’ distribution between male and female were observed (data not shown).

### 3.2. Factors Independently Associated with Death

Factors independently associated with death are shown in Table 3. Overweight (adjusted odds ratio (adjOR) 5.53, 95% CI 2.07–14.76), obesity (adjOR 8.58, CI 3.30–22.29), dyslipidemia (adjOR 10.02, 95% CI 1.06–94.22), heart disease (adjOR 17.68, 95% CI 3.80–82.18), in addition to cancer (adjOR 13.28, 95% CI 4.25–41.51) and male sex (adjOR 5.24, 95% CI 2.30–11.94) were associated with greater risk of death in the youngest population.

Diabetes was not independently associated with increased risk of mortality in the youngest age group but was independently associated with death in older ages (adjOR 1.85, 95% CI 1.05–3.25 and adjOR 1.62, 95% CI 1.04–2.52 in 46–55 and 56–65 age groups, respectively). In addition, hypertension was independently associated with mortality in both groups of older than 45 years of age (adjOR 1.88, 95% CI 1.07–3.32 and adjOR 2.93, 95% CI 1.93–4.45 for the 46–55 and 56–65 age groups, respectively).

### 3.3. Laboratory Parameters at Hospital Admission and before Discharging

Laboratory parameters at hospital admission and before discharging (in both deceased and discharged individuals) are shown in Figure 2. The laboratory profile upon admission to the hospital of survivors is different from that of dead patients. Significantly higher levels of AST (*p* < 0.001), triglycerides (*p* = 0.002), glucose (*p* < 0.001), bilirubin (*p* < 0.001), international normalized ratio (INR) (*p* < 0.01), LDH (*p* < 0.001), CRP (*p* < 0.001), D-dimer and creatinine levels (*p* < 0.001) were observed at both the first and the last measurements in deceased vs. discharged patients. Despite a statistical difference was observed for ALT, being higher at hospital admission for the deceased patients (*p* = 0.045), no difference was observed between the two groups at the last measurement.

## 4. Discussion

The present study highlights that the comorbidity profile associated with COVID-19 mortality is different in younger patients compared with those at elderly age. In this study, we analysed only available data for the consecutive records of dead and alive patients who were admitted to the hospital during the year 2020, in which no COVID-19 vaccine was available and Alpha SARS-CoV2 was the main variant of concern described in Italy. The results of this study based on a population affected by the same predominant SARS-CoV2 variant, homogeneous in terms of lack of effect of vaccination in both deceased and discharged patients, underline better the role of individual risk factors for the worst outcomes of COVID-19 in patients under 65 years of age.

Data from our study, conducted in an Italian population under 65 years of age, suggest that males have a higher risk for death than females and in the oldest population, each of the comorbidities was an independent risk factor for death. Since, in our study, the participants are younger than 65 years, the outcome (death or discharged) differs in relation to age (deceased patients were significantly older than discharged ones), the comorbidity profile was stratified by different age groups. A different independent role of each comorbidity was observed in different age groups. The dysmetabolic profile has a key role in determining the worst outcome in the youngest patients. In particular, among patients between 20 and 45 years, overweight and obesity was significantly correlated with the risk of death, at the highest extent, reaching an adjOR 5–8-times higher compared with the risk observed in patients without these factors. Moreover, in patients older than 45 years, the impaired metabolic profile (diabetes, hypertension and either overweight or obesity) was also independently associated with the risk of death for each factor evaluated. However, in 46–65 years of age patients the adjORs were lower compared with those reported in the youngest patients (varying from 1.49 to 2.36 for patients with overweight and 3.00 to 4.07 for obese patients). As suggested by these data, in accordance with previous Center Disease Control indications, the risk of severe illness from COVID-19 increases sharply with higher BMI [6,13], thus the current concept of only severe obesity (BMI > 40 kg/m^2^) linked to hospitalization and intensive care in hospital should be reassessed. Other comorbidities, which included diabetes, COPD and CKD, were shown to be independently associated with death only in the older and not in the youngest-age patients.

Interestingly, the reported rates of at least one comorbidity related to the metabolic impairment (including diabetes, dyslipidemia, heart disease, hypertension and obesity) were significantly higher among deceased patients compared to those discharged. The same was true also for the presence of three or more comorbidities related to the metabolic impairment: 22.6% vs. 2.2% for deceased and discharged patients, respectively.

It is interesting to note that, among the deceased patients, there is a progressive increase in the number of concomitant comorbidities related to metabolic impairment with increasing age, explained by the fact that with the advancing age, these comorbidities are associated with each other in contributing to the full-blown picture of metabolic syndrome. While 80% of the youngest deceased patients did not have or had only one comorbidity related to metabolic impairment, this percentage dropped to 65% and 44% among 46–55 and 56–65 years of age, respectively. In the youngest patients, the only metabolic comorbidity reported was mainly overweight/obesity.

Our finding that the mortality risk among the youngest patients with severe obesity or with overweight and dyslipidemia was higher than the mortality risk posed by other known comorbidities suggests a significant pathophysiologic link between excess adiposity and dysmetabolic impairment and severe COVID-19 illness. Different studies have tried to evaluate how obesity can play a role in the development of more severe disease in different age groups [6,14]. Kass et al. were among the first to highlight the inverse correlation between age and BMI, reporting that in their dataset of 265 patients, younger individuals admitted to hospital were more frequently obese [8]. Tartof et al. also showed that obesity plays a profound role in risk for COVID-19, particularly in male and younger populations compared to female and older adults [2] and the same was true also for Klang et al. in a cohort of 3406 patients, reporting that among those younger than 50 years of age, BMI ≥ 40 kg/m^2^ was independently associated with mortality (adjOR 5.1), whereas for older patients, the same BMI was also independently associated with mortality to a lesser extent (adjOR 1.6) [15]. In addition, a community-based study analyzing 7 million COVID-19-positive patients confirmed that obesity was strongly associated with mortality in young and middle-aged adult patients [16]. Obesity alone in the outcome of these patients lies in the ability of adipose tissue to impair immune response to viral infection [17], increasing the inflammatory response to viral infections with an increase in oxidant stress that adversely affects cardiovascular function [18,19].

Dyslipidemia, an earlier marker of metabolic impairment, was not a factor independently associated with the risk of death in the overall population, but it was independently associated with death risk only in the youngest age groups. Whether diabetes is an independent factor for severe outcomes remains unclear; however, several studies have shown that diabetes-associated risk was only observed in the younger categories of age [20]. Data from this study have shown that in patients under than 65 years of age, diabetes, as part of a later clinical presentation of a dysmetabolic impairment, was an independent mortality risk factor in patients older than 45 years with an adjOR similar to other comorbidities, but lower than adjOR for overweight, obesity and hypertension at the same age groups. It has been previously reported that the risk of hospitalization for COVID-19 according to age increased from the fifth decade of age in subjects with no diabetes, whereas it increased from 20 to 50 and then reached a plateau in those with type 2 diabetes [20,21]. Previous data and a recent meta-analysis reported that the increased diabetes-related mortality was attenuated in older patients, suggesting that after 50 years of age, the diabetes-related risk is weakened by all other comorbidities or conditions associated with age [22].

Regarding liver disease, as indicated in clinical notes as well as by the Fib4 score, it was significantly higher in deceased versus discharged patients (5% vs. 1%). Zheng et al. showed that the risk of severe COVID-19 illness in patients affected by metabolic-associated fatty liver disease was 5.77-fold greater among obese patients [23]. However, we could only underline the role of the dysmetabolic profile, which frequently affects healthy liver in young patients with the worst outcome of COVID-19. The diagnosis of potential liver damage due to the dysmetabolic profile is underestimated in this study because in severe critical patients, only clinically evident, overt diseases are reported in the clinical notes. In multiorgan failure, at the last stage of COVID-19 illness, liver function tests are frequently altered, without necessarily indicating pre-existing liver damage.

Regarding the biochemical data evaluated, multiorgan failure in the course of a progressive COVID-19 disease in patients who died is confirmed in this study. The surviving patients have lower LDH values (*p* < 0.001) than the deceased patients and this parameter normalized before discharge. This indicates less organ damage attributable to the acute event. Predictably, CRP and D-dimer values are significantly higher in patients who have died, both at hospital admission (*p* < 0.001) and at discharge/death (*p* < 0.001), confirming that increased CRP and D-dimer values are associated with higher mortality. In addition, it is important to underline that in young patients who died due to COVID-19, the values of each biochemical marker are significantly higher in deceased versus alive patients since the first measurement at hospital admission. In this study, it is not possible to suggest a faster progressive disease shortly after infection in patients who died. However, more severe COVID-19 since admission to the hospital and the high-risk profile defined in younger patients could underline the importance of considering, as a fragile population, the subpopulation of young infected adults with impaired dysmetabolic function.

## 5. Limits of the Study

The main strength of the present study is the elevated numbers of younger patients evaluated. Some limits should be underlined to critically evaluate the findings of the study. We were not able to define a potential role for the different medical treatments, which may have influenced outcomes in all hospitalized COVID-19 patients. However, at least during the first wave of COVID-19, all the Italian hospitals used similar treatment protocols, mainly to relieve symptoms. Risk estimates for severe COVID-19-associated illness were measured only among adults admitted to hospital; therefore, these estimates might differ from the risk among all adults with COVID-19. The study population consisted of a nationally representative random sample of deceased patients, 42% of whom (n = 251) were hospitalized in Lombardy Region, while discharged patients included were all from Lombardy Region. To evaluate potential selection bias, we compared patients deceased in Lombardy Region with patients deceased in other Italian Regions and we did not find any significant differences in all the evaluated parameters.

There are no available data on pre-screening effects in the sample population and the data set, thus it was not possible to be evaluated. There is no limitation in the data set; as stated, data doe all consecutive patients are included, though some of the data related to the items collected are not available in some of the clinical notes. Only clinically evident diseases could have been collected in the notes of patients admitted in the hospital in an emergency mode and some comorbidities could have remained undiagnosed or underreported; however, this limit could have impacted both groups of deceased and discharged patients who usually were hospitalized at the same hospital divisions.

Out-of-hospital deaths were not recorded in this study; thus, we cannot exclude that this unavailable information may have affected our results. Another limitation concerns in the lack of BMI in some of the deceased patients.

Furthermore, type of diabetes and its treatment and the duration of diabetes were not reliably indicated, making impossible to determine what type of diabetes is associated with a poor prognosis. In addition, it cannot be excluded that hyperglycemia during hospital admission for COVID-19 might simply be a marker of disease severity and not a modifiable risk factor that can alter outcome. 

The risk factors evaluated in this study have been extensively reported in different populations; however, this specific case-control study design provided interesting data, helpful in increasing the awareness regarding the importance of the metabolic risk in young adults, who should be carefully evaluated for booster vaccination and prompt management of SARS-CoV2 infection.

## 6. Conclusions

While the COVID-19 pandemic will likely remain endemic for several years in the future, strategies for the prevention and treatment of obesity and impaired metabolic health at the population level, focusing also on young populations, need to be promptly put in pace. Young patients who report this risk factor should be better characterized for their risk profile in terms of recommendations to continued booster vaccine prioritization and antiviral therapies, shortly following a symptomatic infection, in order to prevent a severe disease outcome.

## Figures and Tables

**Figure 1 viruses-14-01981-f001:**
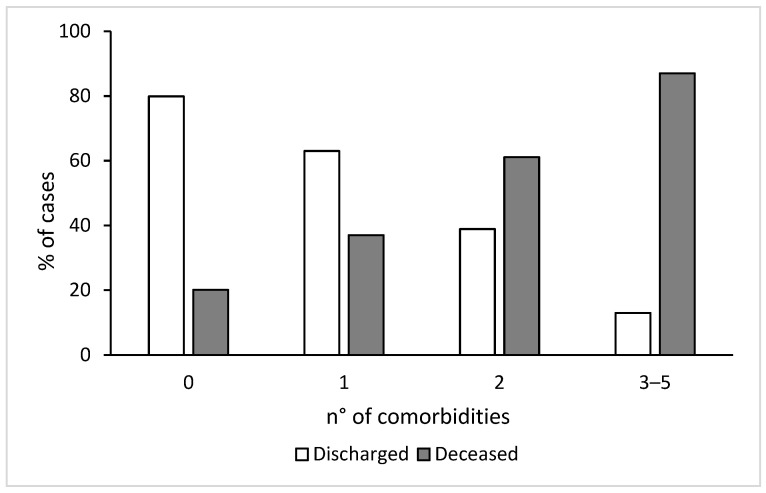
Pattern of comorbidities related to metabolic impairment in discharged vs. deceased patients. Co-morbidities include diabetes, dyslipidemia, heart disease, hypertension and obesity.

**Figure 2 viruses-14-01981-f002:**
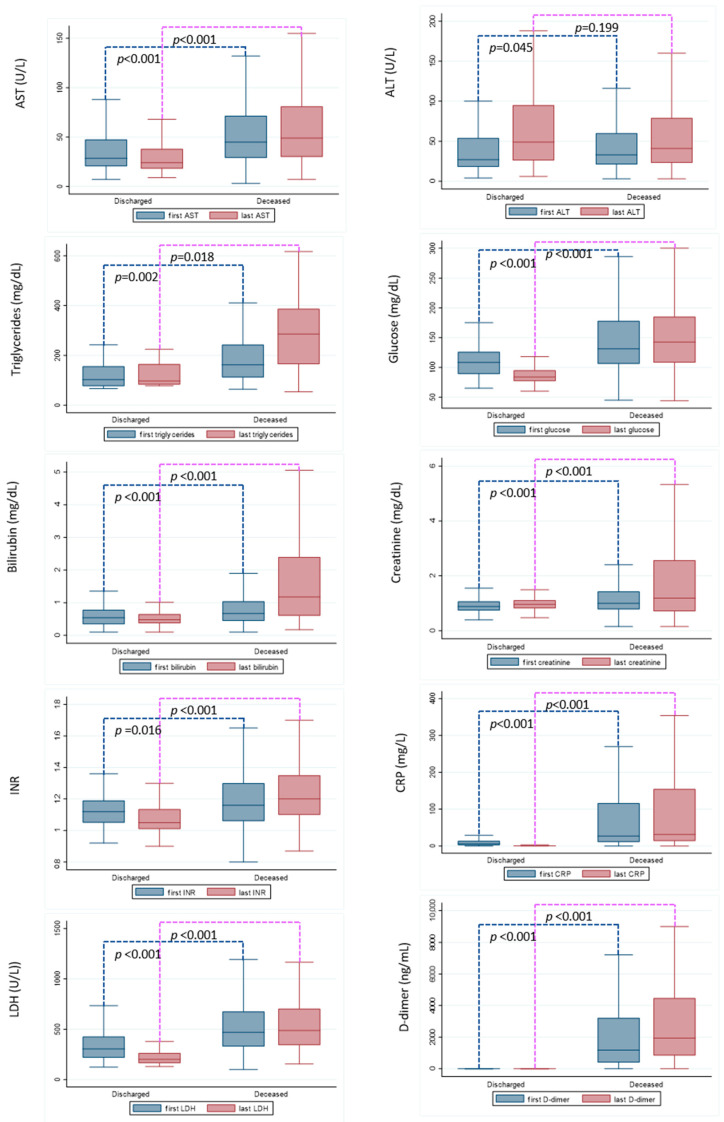
Laboratory parameters at hospital admission and before discharging in deceased vs. discharged patients. Outside values were excluded. The Mann–Whitney U test was used to assess differences between distribution.

**Table 1 viruses-14-01981-t001:** Baseline characteristics of patients included in the study stratified by mortality.

Characteristics	Deceased N = 593 n (%)	Discharged N = 914 n (%)	*p*-Value
**Gender:**			
Males	460 (77.6)	303 (33.1)	<0.001
Females	133 (22.4)	611 (66.9)	
**Age distribution:**			
20–45 year	57 (9.6)	257 (28.1)	<0.001
46–55 years	149 (25.1)	318 (34.8)	
56–65 years	387 (65.3)	339 (37.1)	
**Median Age (range)**	59 (22–65)	52 (20–65)	<0.001
**BMI:**			
Underweight/Normal	190 (32)	603 (66)	<0.001
Overweight	151 (25.5)	179 (19.6)	
Obese	213 (35.9)	130 (14.2)	
Not available	39 (6.6)	2 (0.2)	
**Presence of comorbidities:**			
Diabetes	203 (34.2)	127 (13.9)	<0.001
Dyslipidaemia	62 (10.5)	23 (2.5)	<0.001
Hearth Disease	141 (23.8)	69 (7.5)	<0.001
Hypertension	272 (45.9)	144 (15.7)	<0.001
COPD	67 (11.3)	29 (3.2)	<0.001
CKD	85 (14.3)	44 (4.8)	<0.001
Cancer	88 (14.8)	107 (11.7)	0.077

COPD: chronic obstructive pulmonary disease; CKD: chronic kidney disease.

**Table 2 viruses-14-01981-t002:** Characteristics of study cohort stratified by mortality and age.

	Age 20–45			Age 46–55			Age 56–65		
	Deceased (n = 57; 9.6%)	Discharged (n = 257; 28.1%)	*p*	Deceased (n = 149; 25.1%)	Discharged (n = 318; 34.8%)	*p*	Deceased (n = 387; 65.3%)	Discharged (n = 339; 37.1%)	*p*
**Diabetes**	9 (15.8)	13 (5.1)	0.008	46 (30.9)	49 (15.4)	<0.001	148 (38.2)	65 (19.2)	<0.001
**Dyslipidaemia**	2 (3.5)	3 (1.2)	0.225	7 (4.7)	7 (2.2)	0.153	53 (13.7)	13 (3.8)	<0.001
**Hearth Disease**	9 (15.8)	5 (1.9)	<0.001	24 (16.1)	25 (7.9)	0.007	108 (27.9)	39 (11.5)	<0.001
**Hypertension**	11 (19.3)	21 (8.2)	0.012	50 (33.6)	49 (15.4)	<0.001	211 (54.5)	74 (21.8)	<0.001
**Overweight** **Obese**	13 (25.5) 20 (39.2)	29 (11.3) 32 (12.5)	<0.001	31 (25.3) 48 (36.1)	66 (20.7) 51 (16.0)	<0.001	107 (28.9) 145 (39.2)	84 (24.8) 47 (13.9)	<0.001
**COPD**	3 (5.3)	4 (1.6)	0.086	14 (9.4)	10 (3.1)	0.004	50 (12.9)	15 (4.4)	<0.001
**CKD**	4 (7.0)	9 (3.5)	0.228	16 (10.7)	18 (5.7)	0.049	65 (16.8)	17 (5.0)	<0.001
**Cancer**	11 (19.3)	12 (4.5)	<0.001	27 (18.1)	39 (12.3)	0.090	50 (12.9)	56 (16.5)	0.171
** *N. of comorbidities related to metabolic impairment ** **									
** *0* ** **1** **2** **3 or more**	17 (33.3) 23 (45.1) 5 (9.8) 6 (11.8)	197 (76.9) 45 (17.6) 13 (5.1) 1 (0.4)	<0.001	40 (30.1) 46 (34.6) 27 (20.3) 20 (15.0)	187 (58.8) 94 (29.6) 27 (8.5) 10 (3.1)	<0.001	82 (22.2) 80 (21.6) 100 (27.0) 108 (29.2)	170 (50.3) 115 (34.0) 44 (13.0) 9 (2.7)	<0.001

COPD: chronic obstructive pulmonary disease; CKD: chronic kidney disease. * Including: Diabetes, Dyslipidemia, Hearth Disease, Hypertension and Obesity.

**Table 3 viruses-14-01981-t003:** Multivariable analysis evaluating variables independently associated with mortality in the overall study population and in the 3 age groups.

	Overall (n * = 1466)			Age 20–45 (n * = 307)			Age 46–55 (n * = 451)			Age 56–65 (n * = 708)		
	AdjOR	95% CI	*p*	AdjOR	95% CI	*p*	AdjOR	95% CI	*p*	AdjOR	95% CI	*p*
**Age**	1.05	**1.03–1.07**	**<0.001**	1.05	0.98–1.11	0.138	1.07	0.98–1.17	0.117	1.09	**1.02–1.17**	**0.008**
**Gender** ** *(ref. female)* **	6.92	**5.22–9.17**	**<0.001**	5.24	**2.30–11.94**	**<0.001**	5.41	**3.33–8.79**	**<0.001**	9.00	**6.03–13.45**	**<0.001**
**Overweight** ** *(ref. under-normalweight)* **	2.19	**1.57–3.05**	**<0.001**	5.53	**2.07–14.76**	**0.001**	1.49	0.81–2.72	0.196	2.36	**1.49–3.74**	**<0.001**
**Obese** ** *(ref. under-normalweight)* **	3.81	**2.71–5.36**	**<0.001**	8.58	**3.30–22.30**	**<0.001**	3.00	**1.69–5.34**	**<0.001**	4.08	**2.46–6.77**	**<0.001**
**Diabetes**	1.73	**1.24–2.40**	**0.001**	2.82	0.84–9.48	0.094	1.85	**1.05–3.25**	**0.033**	1.62	**1.04–2.53**	**0.034**
**Dyslipidaemia**	1.70	0.91–3.16	0.096	10.02	**1.07–94.22**	**0.044**	1.32	0.33–5.25	0.691	1.76	0.82–3.77	0.147
**Hearth Disease**	1.73	**1.15–2.61**	**0.009**	17.68	**3.80–82.19**	**<0.001**	1.43	0.66–3.11	0.362	1.44	0.84–2.45	0.183
**Hypertension**	2.41	**1.76–3.30**	**<0.001**	1.24	0.37–4.13	0.726	1.88	**1.07–3.32**	**0.029**	2.93	**1.93–4.45**	**<0.001**
**COPD**	2.50	**1.40–4.48**	**0.002**	1.86	0.20–17.23	0.585	2.22	0.77–6.43	0.142	3.06	**1.40–6.72**	**0.005**
**Cancer**	1.72	**1.18–2.51**	**0.005**	13.28	**4.25–41.51**	**<0.001**	2.05	**1.10–3.83**	**0.023**	1.10	0.65–1.86	0.731
**CKD**	2.33	**1.43–3.80**	**0.001**	0.29	0.06–1.56	0.150	2.62	**1.15–5.97**	**0.022**	3.23	**1.58–6.60**	**0.001**

COPD: chronic obstructive pulmonary disease; CKD: chronic kidney disease. * Number of cases with complete information for each variable included in the model. Age was introduced as continuous as the effect of age on COVID-19 lethality is linear, i.e., the OR = 1.05 means that the risk of death is 5% more for each year of age. OR = 1 is the “reference” risk for people aging as the youngest case of the group under analysis which is 46 years old, when analysing the age group 46–55. Bold numbers indicated statistically significant values.

## Data Availability

Cumulative data are reported within the paper, whereas each patient’s data are not fully available for ethical reasons. The SARS-CoV2-positive deaths surveillance Group (Istituto Superiore di Sanità; cinzia.lonoce@iss.it) and Massimo Puoti (ASST Grande Ospedale Metropolitano Niguarda; massimo.puoti@ospedaleniguarda.it) are in charge for data management and readers may contact them for specific data requests. They will provide the necessary ethical clearances for access to data.

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
