# Peer review of "Obesity and Dysmetabolic Factors among Deceased COVID-19 Adults under 65 Years of Age in Italy: A Retrospective Case-Control Study"

_viruses, 2022, doi:10.3390/v14091981_

Round 1
Reviewer 1 Report
Dr. Kondili et al. conducted a retrospective analysis on the Italian patients affected by COVID-19 during February-December 2020. The work is well structured, organized and narrowly focused. The Authors analyzed the prognostic value of obesity and dysmetabolic factors on mortality risk in hospitalized patients with less than 65 years. Several confounding factors might have impacted on a possibile associationism, however the Authors admitted these limitations.
As minor points:
- the novelty is a weakness since literature has already ascertained these findings by now.
- in the methods section, non parametric tests were mentioned, therefore I assumed non-linearity of the data. I wonder whether a multivariate logistic regression represents the most appropriate approach or rather a generalized linear model, robust regression, quantile regression should be preferred. The Authors ought to be defend more in-depth their choice and specify in detail when the data distribution was normal or not (which test were performed to assess so?). These reflections may be useful to avoid overdispersion of the data.
Author Response
Point 1: The novelty is a weakness since literature has already ascertained these findings by now.
Response 1: We agree that the risk factors evaluated in this study have been extensively reported in different populations, however we believe that this specific case-control study design provide interesting data helpful to increase the awareness on the importance of the metabolic risk also in the young adults. The population with metabolic risk factors should be carefully evaluated for booster vaccination and prompt management of SARS-CoV-2 infection. This is now better clarified in Discussion section (L451-455).
Point 2: In the methods section, non parametric tests were mentioned, therefore I assumed non-linearity of the data. I wonder whether a multivariate logistic regression represents the most appropriate approach or rather a generalized linear model, robust regression, quantile regression should be preferred. The Authors ought to be defend more in-depth their choice and specify in detail when the data distribution was normal or not (which test were performed to assess so?). These reflections may be useful to avoid overdispersion of the data.
Response 2: Thank you for this comment. The authors choose to utilize non-parametric statistics because in many cases, especially in table 2, the number of subjects in each category is quite small; also, a normality test could be affected by this and could result in a “non-significant deviation from normality” just because of lack of power. As the statistical significance has been achieved for most comparisons, also by a non-parametrical less-powered test, the authors preferred not to “risk” biased results, associated with the use of a parametric test, when normality is not ensured. Regarding the multivariate analysis, we preferred a logistic regression, instead of a generalized model, because it allows to directly compute OR; in fact, we believe that the strength of the association represented by OR is very simple to understand for the readers even though they do not have particular statistical expertise.
To clarify our choice in the statistical methods used we added the following comments in the method section of the revised version of the manuscript:
“Non-parametrical less powered tests have been used to be more conservative in defying the statistical significance in particular in lack of power when the normality is non ensured.” (L112-113)
“Regarding the use of the variable “age”, in each model it was introduced as continuous as the effect of age on COVID-19 lethality is linear.” (L119-120)
Reviewer 2 Report
This study, analyzed the existed medical data, is relatively simple but also significantly important for the prevention and treatment of the COVID-19 outbreak in this stage. In the analysis through statistical methods, however, the correct use and logical deduction were most important to the studies like this. Some issues and concerns of mine are as follows.
In addition to the differences in the individual physical conditions of the patients themselves, in practice, vaccines have been identified as one of the important factors in the diagnosis and even death of COVID-19. The previous studies also confirmed this point. However, why vaccination status of the sample population did not include in this study? In the database analyzed in this study, is there no record of vaccination of patients? Moreover, different strains of COVID-19 could affect its infectivity and diagnosis rate, so background information in this part of the study becomes important, such as which strain of the patient samples were infected in the surveys, that is, Alpha, Delta, or Omicron?
Title, Obesity and dysmetabolic factors among deceased COVID‐19 young adults in Italy: a retrospective case‐control study, I do not think the term “young adult ” is appropriate to exactly present the fact of the study which the sample population examined was younger than 65 years. Thus, how to define the young adults?
P47, If the authors said several reports, it had better to cite more than one reference.
L62, Study population and data collection, more information is needed. For example, the data set is an open access? Or need to charge? How about the data format and all cases happened in Italy are included? Was any missing data? Any limitation for the data set?
L64, Please spell out the term ISS when it was mentioned first time.
L101-102, please merge this sentence into the previous paragraph.
L111, IQR, Please spell out the term.
Figure 1 can be combined as one figure to show the trend when the percentage of patient without any comorbidity, with one, two and three or more than 3 comorbidities.
Please note that Table 3 was not mentioned in the text, and it may be also very confusing to readers what did this table mean. The table should provide more information, such as significance level and observation number, etc. By the way, the table is not clearly stated. For example, age variable, what’s the cut-off point? Because the analysis was based on all data; if OR=1.05, what did this mean?Otherwise another OR should be 1, thus, 1.05 and 1 represent what age levels respectively?
L215-216, “…varying from 1.49-2.36 risk for patients with over- 215 weight and 3.00-4.07 for obese patients of 46-65 years of age.” Unclear sentence. What did these numbers mean?
Some term “data not shown” appeared in the text may be inappropriate in a technical report. No data, no evidence.
L320, Conclusion should be conducted as an independent section.
Study limitation, please check if any pre-screening effect existing in the sample population and the data set.
Author Response
Point 1: In addition to the differences in the individual physical conditions of the patients themselves, in practice, vaccines have been identified as one of the important factors in the diagnosis and even death of COVID-19. The previous studies also confirmed this point. However, why vaccination status of the sample population did not include in this study? In the database analyzed in this study, is there no record of vaccination of patients? Moreover, different strains of COVID-19 could affect its infectivity and diagnosis rate, so background information in this part of the study becomes important, such as which strain of the patient samples were infected in the surveys, that is, Alpha, Delta, or Omicron?
Response 1: Thank you for this comment which gives us the possibility to better clarify the study population. We totally agree that different strains of COVID-19 affect its infectivity, diagnosis rate and also the vaccination has an important role in the outcome of SARS-CoV-2 infection. We better clarified that in this study were analysed only available data of the consecutive records of death and alive patients who were admitted to the hospital during the year 2020, time in which no COVID-19-vaccine was available and Alpha was the main variant of concern described in Italy. We believe that the population affected by the same predominant SARS-CoV-2 variant and the homogeneity in terms of lack of effect of vaccination in both died and survived discharged patients make the results more meaningful in terms of the role of individual risk factors for important outcomes of COVID disease to be addressed for their future better management in terms of booster vaccination and prompt antiviral therapy in infected patients. (L229-236)
Point 2: Title, Obesity and dysmetabolic factors among deceased COVID‐19 young adults in Italy: a retrospective case‐control study, I do not think the term “young adult” is appropriate to exactly present the fact of the study which the sample population examined was younger than 65 years. Thus, how to define the young adults?
Response 2: We specified in the title the study population of adults under 65 years of age
Point 3: P47, If the authors said several reports, it had better to cite more than one reference.
Response 3: Thank you, the phrase has been corrected as suggested.
Point 4: L62, Study population and data collection, more information is needed. For example, the data set is an open access? Or need to charge? How about the data format and all cases happened in Italy are included? Was any missing data? Any limitation for the data set?
Response 4: As suggested, more information on study population and data collection have been included in the Methods section (L68-72; L94-95). There is no limitation in the data set; as stated data of all consecutive patients are included, however some of the data related to the items collected are not available in each of clinical notes and this is acknowledged in the limitations of the study (L391-393).
Point 5: L64, Please spell out the term ISS when it was mentioned first time.
Response 5: Thank you, the term ISS has been spelled out the first time mentioned.
Point 6: L101-102, please merge this sentence into the previous paragraph.
Response 6: As suggested, the sentence (L120-121 of the revised version) has been merged with the previous paragraph.
Point 7: L111, IQR, Please spell out the term.
Response 7: The term IQR has been spelled out. (L130)
Point 8: Figure 1 can be combined as one figure to show the trend when the percentage of patient without any comorbidity, with one, two and three or more than 3 comorbidities.
Response 8: The Figure 1 has been revised as suggested
Point 9: Please note that Table 3 was not mentioned in the text, and it may be also very confusing to readers what did this table mean. The table should provide more information, such as significance level and observation number, etc. By the way, the table is not clearly stated. For example, age variable, what’s the cut-off point? Because the analysis was based on all data; if OR=1.05, what did this mean? Otherwise, another OR should be 1, thus, 1.05 and 1 represent what age levels respectively?
Response 9: Mention to table 3 has now been added in the Results section (L191). In addition, more information has been added to table 3; in particular, n and p-values have been introduced.
Regarding the use of the variable “age”, in each model it was introduced as continuous as the effect of age on COVID-19 lethality is linear. The OR=1.05 means that the risk of death is 5% more for each year of age. OR=1 is the “reference” risk for people aging as the younger case present in the group under analysis. This explanation has been added for clarity in the revised version of the manuscript as footnote of Table 3. (L207-211)
Point 10: L215-216, “…varying from 1.49-2.36 risk for patients with overweight and 3.00-4.07 for obese patients of 46-65 years of age.” Unclear sentence. What did these numbers mean?
Response 10: Thank you for this comment. The text was rephrased in order to increase the clarity. (L249-251)
Point 11: Some term “data not shown” appeared in the text may be inappropriate in a technical report. No data, no evidence.
Response 11: Comments on “Data not shown” have been deleted
Point 12: L320, Conclusion should be conducted as an independent section.
Response 12: The Conclusion paragraph has been conducted as an independent section
Point 13: Study limitation, please check if any pre-screening effect existing in the sample population and the data set.
Response 13: Unfortunately, there are no data on pre-screening effect in the sample population and the data set and because it is not evaluated and was not the aim of this study we are not recognizing it as a limitation of the study.
Round 2
Reviewer 2 Report
I appreciate all responses by the authors; the revision work is pretty good. Specifically, the concerns regarding the vaccination and strains of COVID-19 were crucial for the study. When reporting “We believe that the population affected by the same predominant SARS-CoV-2 variant and the homogeneity in terms of lack of effect of vaccination in both died and survived discharged patients make the results more meaningful in terms of the role of individual risk factors for important outcomes of COVID disease to be addressed for their future better management in terms of booster vaccination and prompt antiviral therapy in infected patients." I totally agree with this viewpoint. Since the study results are solid, it is hoped that this study can contribute to the prevention and treatment of COVID-19 epidemic.